# The Problem of Monitoring Activities of Older People in Multi-Resident Scenarios: An Innovative and Non-Invasive Measurement System Based on Wearables and PIR Sensors

**DOI:** 10.3390/s22093472

**Published:** 2022-05-03

**Authors:** Riccardo Naccarelli, Sara Casaccia, Gian Marco Revel

**Affiliations:** Department of Industrial Engineering and Mathematical Sciences, Polytechnic University of Marche, 60131 Ancona, Italy; s.casaccia@staff.univpm.it (S.C.); gm.revel@staff.univpm.it (G.M.R.)

**Keywords:** multi-resident activity monitoring, smart home, activity level, older people, PIR sensor, BLE, wearable tag

## Abstract

This paper presents an innovative multi-resident activity detection sensor network that uses the Bluetooth Low Energy (BLE) signal emitted by tags worn by residents and passive infrared (PIR) motion sensors deployed in the house to locate residents and monitor their activities. This measurement system solves the problem of monitoring older people and measuring their activities in multi-resident scenarios. Metrics are defined to analyze and interpret the collected data to understand daily habits and measure the activity level (AL) of older people. The accuracy of the system in detecting movements and discriminating residents is measured. As the sensor-to-person distance increases, the system decreases its ability to detect small movements, while still being able to detect large ones. The accuracy in discriminating the identity of residents can be improved by up to 96% using the Decision Tree (DT) classifier. The effectiveness of the measurement system is demonstrated in a real multi-resident scenario where two older people are monitored during their daily life. The collected data are processed, obtaining the AL and habits of the older people to assess their behavior.

## 1. Introduction

With the development of the Internet of Things (IoT) and the rise of powerful and affordable smart devices, sensing technologies have become the basic pillars of any smart application today to provide independent living solutions for aging people. Monitoring the practices, activities and habits of older people is essential to assess their health and ability to live independently [1,2]. Automatically monitoring the activities of a single resident in a smart home is relatively a simple task, while monitoring multiple residents remains an open challenge. Thus, applications are needed to recognize the activities and behaviors of multiple residents in the same environment. This is a challenging task because unlike the single-resident case where the sensor states directly reflect the activity of a specific person, in the case of multi-resident scenarios that information is generally not known, so it becomes difficult to associate activities with the people who carry them out. For example, if we assume that two motion sensors are activated at the same time in two different rooms, although this indicates the presence of two people, we do not know which person is in which room. Therefore, the complexity of activity recognition emerges and data binding (associating the collected data with the resident who caused them) becomes essential: the key challenge in recognizing residents’ activities is to associate sensors’ observations with residents’ identities.

The most used solutions for smart homes to track residents while carrying out activities are:-Vision-based [3,4,5,6,7]: they use biometric information (e.g., face, height) extracted from camera videos to recognize residents as they go about their daily activities. The main issue in this approach is privacy.-Tag-based [8,9,10,11,12,13,14]: they use portable devices that provide a unique identifier (ID) to the wearer. These devices use different communication technologies such as Wi-Fi, Radio Frequency Identification (RFID), Bluetooth Low Energy (BLE) and ZigBee. Scanners placed in different areas of the house that listen for broadcast packets are needed to locate the tags and the individuals.-Signature-based [12,15,16,17,18,19,20]: they use different types of sensors such as microphones, passive infrared (PIR) motion sensors, ultrasound, and ultra-wideband (UWB) to generate a unique signature for the resident. The main drawback of this approach is the accuracy and dependence on the environment in which it is used.

Many approaches such as data association [21,22] and artificial intelligence (AI) algorithms [23,24,25,26,27] are used to monitor older people’ activities and behaviors in multi-resident scenarios. Sensors installed in the home can capture useful information when older people go about their daily routine: activities and movements are tracked to give an overview of their functional status to improve it when necessary [28,29]. Staying active is essential for older people to prevent disease, maintain independence and improve their quality of life. Studies that have monitored the activities of older people use many types of technology-based monitoring techniques for recording daily activities [30,31]. These include the use of PIR sensors [32,33,34,35,36,37], cameras [38,39,40], pedometers [41,42,43] and accelerometers [44,45,46,47]. Pedometers are widely used to record activity, but the type of information derived is limited to just the collected step count and does not include the gait pattern. Furthermore, pedometers often underestimate the number of steps during a slower or irregular/unstable gait seen in older people [48]. The limits of pedometers have been overcome by accelerometers, which can monitor activity by providing its frequency and intensity (light, moderate, vigorous) through raw acceleration signals [49,50]. However, the data-processing algorithms are often proprietary, so we do not have access to the criteria by which activities are classified; although in some cases the sensors allow access to raw data, these require special programming to calculate the intensity of the activity of older people [51]. Although the use of body-worn devices is versatile in providing continuous monitoring even outdoors, PIR sensors and cameras free residents from wearing devices in their homes, but, while cameras arise privacy concerns, the PIR sensors are well accepted [52]. In [53], the authors process the data collected by PIR sensors to map the daily activities of the home resident: mobility-related features and duration of specific activities are determined to assess behavior patterns of the resident. The authors of [54] present a methodology of analyzing data from PIR sensors recorded in older people’s homes to acquire activity patterns representing different health conditions. These data are displayed in the form of a density map showing different levels of activity. Three case studies of older people are included to show how the density map and measure of dissimilarity can be used to track general activity levels and daily patterns over time, showing changes in health status. Detecting activity levels and behavior changes over a long period of monitoring is useful for assessing the functional health of older people [55,56].

The main contribution of this paper is the presentation of an innovative multi-resident activity detection sensor network that combines tag-based (wearable BLE tags) and signature-based (PIR motion sensors) technologies to locate and identify residents and monitor their activities in multi-resident scenarios. The ease of use and deployability of the sensor network, its low cost, non-invasiveness, respect for privacy, robustness and reliability are all advantages that make it suitable for installation in the home to monitor residents compared to systems in the literature. Furthermore, this system can be extended to larger environments and contexts with more residents. The sensor network can also be used in nursing homes where the monitoring of older residents can help caregivers understand their habits and daily activities. The system can track and identify each resident wearing a BLE tag, achieving 96% accuracy in identifying residents using the DT classifier. The effectiveness of the measurement system is demonstrated in a real multi-resident scenario where two older people are monitored simultaneously during their daily life and their activity levels are discriminated. Finally, we show that collected data can be processed by defining specific metrics to understand daily habits and measure the Activity Level (AL) of older people. The methodology proves to be an innovative approach to define a behavioral model: from the number of detected events (PIR activations) the parameters relating to the activity of the older people are obtained, while from the number of events acquired per location, it is possible to understand how much activity they perform to accomplish Activities of Daily Living (ADL). The assessment of this model can help to capture possible changes resulting from cognitive or physical health and to prevent or prognosticate diseases. Early detection of behavioral changes with the assistance of this technology can help improve health interventions and provide older people with greater awareness of an unhealthy and inactive lifestyle.

The remaining part of the paper is organized as follows: Section 2 provides an overview of the literature on sensing technologies for multi-resident scenarios, Section 3 presents the multi-resident activity detection sensor network and data collection approach, Section 4 defines metrics for data processing to assess older people’s AL and habits, Section 5 describes the testing phases, Section 6 presents the results of the tests and the analysis of the collected data, Section 7 discusses the results, and finally, Section 8 describes the main conclusions of the paper.

## 2. Related Work

Distinguishing the residents of the house is the first step in associating the detected activities with each of them. For this purpose, different technologies are used to continuously locate and identify residents in the home. As reported in Section 1, there are three main solutions to distinguish the home inhabitants: vision-based approaches, tag-based approaches and signature-based approaches. Vision-based approaches capture images by a camera to use them to identify residents. Although they suffer from background changes, illumination variations and environmental noise, high accuracy is achieved in many cases. The use of cameras also raises the problem of privacy, so it is considered an invasive approach and, in most cases, it is not accepted. In addition, the high cost of hardware and deployment reduces the use of vision-based technology. Tag-based approaches use wearable devices that provide a unique ID to the owner. These devices send broadcast packets via different communication technologies to scanners located in the house. In this way, devices and therefore individuals are located. This technology has the advantage of being portable, low in cost, reliable, non-invasive, and respectful of privacy. The main drawbacks are the need to have many antennas/scanners in the house, the possible fluctuation of the signal due to the presence of external factors such as walls, objects, etc. (generate absorption, interference, diffraction), and the possibility that the resident forgets to wear the device. The signature-based approach instead frees residents from having to wear devices, relying on data collected from various types of sensors installed in the environment. These sensors can provide awareness of resident’s context (location, activities) and ensure privacy. However, it is necessary to install sensors in the home for a long period of time, which can lead to installation and maintenance costs. This approach is highly dependent on the environment in which the sensors are installed, which directly affects accuracy.

Parameters such as residents’ comfort and privacy, their role in data collection (i.e., wearing a device), the accuracy of the technology, its cost, installation and invasiveness have a significant impact on their experience and are all involved in the decision on the sensing system choice. Since the use of vision-based solutions raises the issue of privacy, and since it is nearly impossible to provide data-binding from signature-based sensors alone, tag-based technologies are the most promising in recognizing residents. In this Section, we present literature works dealing with tag-based technologies that proved to be a promising solution for the purpose of locating and identifying home residents.

Tag-based approaches rely on the use of wearable wireless technology to locate and identify people in the home. Each device has a unique ID that can be broadcasted to a receiving unit via wireless communications as BLE, Wi-Fi, RFID or ZigBee. In [57] a Wi-Fi-based system which relies on the use of a smartphone is proposed for indoor localization reaching an average accuracy of 2 m. The approach used in [58] exploits the combination of movement detection in a multi-resident context based on motion sensors and smartphones. The results show that this approach improves the accuracy of ADL classification by more than 30% compared to smartphone-only solutions. However, systems using Wi-Fi technology require many antennas and a fingerprint phase. Moreover, since smartphones are almost the only wearable devices equipped with Wi-Fi technology, each resident must have one to be tracked and identified in the home. The authors of [59] present a room-level localization and identification system based on the analysis of RSSI for ZigBee wireless sensor networks. Even if the system achieved 98% accuracy, there is the disadvantage of using a specifically designed ZigBee portable device, because there are no commonly used devices equipped with this technology. Furthermore, the system requires a time-consuming fingerprint phase, which makes it impractical for our purpose. In [60,61] localization systems based on RFID are developed, achieving sub-centimeter accuracy. The drawbacks of using this communication technology are the high cost (RFID reader and antennas are expensive) and that it is not used in wearable devices. Finally, many authors presented promising accurate indoor positioning and identifying systems based on BLE technology. In the multi-resident smart home scenario of [62], the individuals wear a BLE tag that conveys their ID at certain time intervals while stationary scanners listen to the packet of broadcast to locate them. However, the accuracy of the localization is highly dependent on the Received Signal Strength Indication (RSSI) and the time interval of emission of the BLE tags. To achieve the best localization accuracy, they developed an algorithm for selecting the optimal value settings of the BLE sensors. Residents in [63] are provided with an iBeacon bracelet that transmits data packets that are collected using a network of devices that act as receiving antennas. The system evaluates the path loss of the received signal and corrects the calculated position with a probabilistic approach to avoid wall-crossing. Location data are combined with information from other IoT devices to analyze behaviors, habits, and social interactions among patients. The system proposed by [64] estimates the areas in which residents are located with a BLE beacon, using various methods to analyze the RSSI values of the BLE signals and recognize the activity of each resident from the estimated area information. The information on the residents’ activity is then used for the creation of daily reports. The authors of [65] design, implement and evaluate an IoT-based system that uses BLE technology for indoor localization. Even if it can estimate the position with good accuracy, it requires a long training process and several machine-learning algorithms, making it useless in large environments. In [66], it is shown that, by making people inside the house wear a BLE bracelet, it is possible to locate them with an accuracy of 90.22%. By adding motion detection to the localization algorithm, accuracy is increased to 92.98%. The authors in [67] propose a monitoring system for the older people able to monitor the activities of several people simultaneously by estimating the probable location area of each of them. The system, consisting of mobile beacons and a fixed scanner, estimates the probable location area of several people with high precision using the RSSI of BLE devices.

As can be seen from the literature, the monitoring of multiple residents in the same indoor environment remains an open challenge, which researchers are trying to answer with different technologies and approaches. In the case of older people-related monitoring systems, it is recommended to use a solution with high accuracy, low cost, low energy consumption and reliability. Based on these requirements, BLE devices are promising technologies. Furthermore, we found that combining tag and signature-based technologies could result in improved activity recognition.

## 3. Materials and Methods

### 3.1. System Architecture and Sensor Integration

To monitor older people living in the home, it is necessary to associate the detected activities with each resident. To achieve this goal, we present a multi-resident activity detection sensor network that makes use of both signature-based and tag-based technologies. The components that make up the system have been chosen based on some requirements: they had to be inexpensive, non-invasive, respectful of privacy, with long battery life, portable and not bulky. We selected the HC-SR501 PIR sensor (technical features reported in Table 1) as the motion detection sensor, the Tile Mate BLE tag (technical features reported in Table 2) as the individual identification technology, and the Raspberry Pi Zero W (technical features reported in Table 3) to act as the processing unit and BLE signal scanner.

The Raspberry Pi Zero W is one of the best nano-computers in terms of size (65 mm long by 30 mm wide), price, performance, memory and connectivity. Furthermore, due to the availability of Wi-Fi and Bluetooth communication technologies, the Raspberry Pi Zero W is ideal for making embedded IoT projects. The PIR motion sensor emits a pulse when there is a significant change in the amount of infrared radiation it detects. There are three pins on the PIR sensor: one for the power (Vcc), one for the ground (Gnd) and one for the transmission of the signal of the detected movement (Out). The PIR sensor is wired to the Raspberry Pi Zero W (Figure 1): the Vcc pin of the PIR sensor needs to be connected to a 5 V pin on the Raspberry Pi Zero W; the Gnd pin can be connected to any ground pin on the Raspberry Pi Zero W; lastly, the Out pin needs to be connected to any of the GPIO (general purpose input/output) pins on the Raspberry Pi Zero W.

The Tile Mate (Figure 2) tags are easily worn around the neck by home residents throughout the day thanks to their compact size.

The multi-resident activity detection sensor network is designed to ensure simultaneous monitoring of two residents in three areas of the house; therefore, it consists of three PIR motion sensors wired to three Raspberry Pi Zero W devices and two wearable tags. Each Raspberry Pi Zero W is mains-powered and is equipped with a micro-SD card on which the Raspbian OS runs. Considering the cost of individual components, power supplies, electronics and storage, the proposed system costs around EUR 200. An advantage of this system is that it can be easily extended to larger environments and contexts with more residents. In fact, by integrating the system with more Raspberry Pi Zero W devices and PIR motion sensors it can cover more areas of the house (based on the number of rooms in which we want to monitor the residents, we increase the number of PIR sensors and Raspberry Pi Zero W scanners), while integrating it with additional tags, we can track more people (each resident is assigned a tag and associated with a unique ID).

### 3.2. Resident Detection and Identification

The sensor network is designed to be installed in three main rooms of the home. Thanks to the integrated Wi-Fi module, the Raspberry Pi Zero W devices are connected to the home internet network, so that they store the data collected by the own PIR motion sensors in an online database. A Google spreadsheet was used as the IoT database to store sensor data and retrieve them for further processing. Through the Google Sheets API, the online spreadsheet access credentials and JSON security key have been created. The OAuth 2.0 authorization protocol was used to allow authentication and access to the Raspberry Pi Zero W devices via the JSON key for sending the collected data and saving them in the Google spreadsheet.

The multi-resident activity detection sensor network (Figure 3) works following six main steps:Each resident of the home wears a tag that identifies them through a unique identifier (ID), which consists of the MAC address of the tag.When a PIR motion sensor is triggered by a resident’s movement, the Raspberry Pi Zero W connected to it scans the surrounding area searching for devices that emit BLE signals: the system is set to detect only the BLE signal of tags worn by residents.The BLE RSSI of the tags in the detection area are measured.To identify the resident who activated the PIR sensor, the measured RSSIs are converted into distances (in meters). The presence in the room is thus assigned to the residents in the detection area of the PIR sensor.

Combining the Measured Power (*MP*), a factory-calibrated read-only constant that indicates the expected *RSSI* at a distance of 1 m to the tag, with the measured *RSSI*, the distance of the tag (i.e., the resident) from the PIR sensor is estimated. For the Tile Mate tag, the *MP* is −58 dBm.

The distance is computed through Equation (1) if $\frac{RSSI}{MP}<1.0$:(1)$$\mathrm{Distance}={\left(\frac{RSSI}{MP}\right)}^{10},$$
while it is computed through Equation (2) if $\frac{RSSI}{MP}\geq$1.0:(2)$$\mathrm{Distance}=0.89976\times {\left(\frac{RSSI}{MP}\right)}^{7.7095}+0.111$$

5.The system, meanwhile, stores the collected data in real-time in the online Google spreadsheet for further analysis. Each time the system detects a movement, it stores the identity of the resident (anonymously called “Resident 1” and “Resident 2”), the time of the event, and its location (the room in the house where motion is detected).6.These data are then processed to obtain residents’ AL and habits using the metrics defined in Section 4.

### 3.3. PIR Sensors Deployment

The sensor network, in particular the PIR motion sensors, must be installed in such a way as to monitor the residents at least in the areas of interest within the smart home environment according to a pre-selection of which areas must be covered. In the real multi-resident scenario test, a PIR sensor is installed for each room where residents are expected to spend most of their time, i.e., the kitchen, bathroom and bedroom, in order to monitor their activities.

To optimize data collection, we have listed several recommendations for PIR sensor deployment in the smart home that come from our experience:Once the rooms of the house in which to monitor the residents have been identified, the PIR sensors should be installed in such a way as to cover the entire area. Therefore, it is important to know, in addition to the detection range of the PIR sensor, the dimensions of each room to evaluate the number of sensors needed to cover them.PIR sensors should be installed avoiding overlapping detection areas to prevent more than one sensor from detecting the movement of a resident, thus creating false events.It is preferable to install the PIR sensors at a minimum height of 1.5 m from the ground to better detect residents’ movements.It is preferable to avoid installing the PIR sensors near doors or leaves so that they are not obscured by them.

## 4. Activity Level and Habits Assessment

Behavioral changes can reflect changes in physical health and the continuous collection of sensor data can help observe these changes, which in some cases may be difficult to detect. Understanding daily habits and monitoring the AL of older people are essential to assess their health and ability to live independently. Therefore, we defined some metrics to analyze and interpret the data collected by the proposed system to extract useful information.

Section 4.1 and Section 4.2 show the parameters relating to daily activity and habits extracted from the dataset, respectively.

### 4.1. Daily Activity Level

The number of detected events (PIR activations) per day may reflect how physically active the older resident is. To assess the AL of older people, three ALs named Low Activity Level (LAL), Medium Activity Level (MAL), and High Activity Level (HAL) have been set. Considering that the minimum AL for healthy older people is reflected in 50 events per hour [9] and from observations we made in the experimental phase with real older subjects, we set the LAL below 1000 events per day, the MAL between 1000 and 2000 events per day, and the HAL at more than 2000 events per day.

The Daily Activity Index (DAI) reflects the activity performed by older people during the whole day. So, the resident DAI assumes a value of 1 if the performed activity falls within the LAL, 2 if the performed activity falls within the MAL, and 3 if the performed activity falls within the HAL.

### 4.2. Daily Habits

One of the problems of interest is to monitor the habits of older people. Residents carry out their activities at different times of the day, so it is useful to measure how often they perform activities on an hourly scale. The Hourly Activity Percentage (HAP) is computed through Equation (3) by dividing the number of events detected per hour in the whole house by the total daily events and allows determination of the hours of the day when residents are more active.
(3)$$\mathrm{HAP}\;\left[\%\right]=\frac{\sum^{\;}Hourly\;events\;}{Total\;daily\;events}\times 100\%$$

Furthermore, the collected movements of older people inside the home can be used to retrieve the activities carried out in each room. The multi-resident activity detection sensor network is designed initially to be installed in three rooms of the house: bathroom, bedroom, and kitchen. The number of events captured per location may reflect how many of older people’s activities are assigned to perform a specific ADL such as personal hygiene, eating, etc. The Daily Room Activity (DRA) of each resident is computed by dividing the number of events detected in a room by the total number of events for each day (Equation (4)).
(4)$$\mathrm{DRA}\;\left[\%\right]=\frac{\sum^{\;}Room\;events\;}{Total\;daily\;events}\times 100\%$$

Identifying typical habits can help capture possible changes in older people’s behavior by comparing two or more periods of collected data. If a noticeable difference emerges, it can be considered a symptom of an impending decline in physical and cognitive abilities and requires further investigation.

## 5. Test Phases

The test phases can be divided into two steps. First, the measuring capabilities of the multi-resident activity detection sensor network were evaluated. We determined the accuracy of the system in detecting different types of movements (e.g., walking, moving the arms, standing movements) and its accuracy in discriminating the residents. We then moved on to installing the system inside a home inhabited by two older people to evaluate its functioning in a real multi-resident scenario and the effectiveness of the proposed metrics in determining the AL and habits of the older residents.

### 5.1. Movement Detection

To measure the accuracy of movement detection, a test was performed by having an individual wearing the tag positioned at a given distance from the PIR sensor. The participant was asked first to make small movements (turning the head or moving the hands) and then to make large movements (moving the arms/legs or walking in place) while remaining in the same spot. The procedure was repeated 20 times for each of the two different types of motion (small vs. large) for 7 different distances from the PIR sensor, to define the accuracy of the system in detecting different types of movements at different distances. The accuracy is computed through Equation (5):(5)$$\mathrm{Accuracy}\;\left[\%\right]=\frac{Number\;of\;movements\;detected}{Total\;number\;of\;movements}\times 100\%$$

### 5.2. Residents Discrimination

Two individuals wearing the tags performed a supervised test designed to measure the accuracy (Equation (6)) of the system in discriminating against residents. While one participant remains still, the other moves into the detection range of the PIR sensor to activate it. The process is repeated alternating between the participant who moves around the room and the one who remains motionless. For each of the participants, 250 PIR sensor activation events with relative ID detection were recorded for a total of 500 events. A third person tracks the movements of the participants to provide a reference for the comparison between the identity of the participant declared by the measurement system and the actual one. The identified events are then labelled as correct or incorrect detections based on whether the ID declared by the system matched the supervisor-defined ground truth or not.

Furthermore, after determining the accuracy of the system, we tested some Machine Learning (ML) algorithms on the collected dataset to assess whether they would increase the effectiveness in assigning the detected activity to the correct individual. Four supervised ML algorithms commonly used for classification problems were tested on the dataset: Decision Trees (DT), Support Vector Machine (SVM), k-Nearest Neighbors (k-NN), and Naive Bayes (NB). Accuracy (Equation (6)), precision (Equation (7)), recall (Equation (8)), and F1-score (Equation (9)) were calculated by splitting the dataset into 70% for training and 30% for testing. Furthermore, DT, SVM, k-NN and NB accuracies were calculated by splitting the dataset using 10-Fold Cross Validation (CV).
(6)$$\mathrm{Accuracy}\;\left[\%\right]=\frac{\mathrm{TP}+\mathrm{TN}}{\mathrm{TP}+\mathrm{TN}+\mathrm{FP}+\mathrm{FN}}\times 100\%$$
(7)$$\mathrm{Precision}\;\left[\%\right]=\frac{\mathrm{TP}}{\mathrm{TP}+\mathrm{FP}}\times 100\%$$
(8)$$\mathrm{Recall}\;\left[\%\right]=\frac{\mathrm{TP}}{\mathrm{TP}+\mathrm{FN}}\times 100\%$$
(9)$$F1\;\mathrm{score}\;\left[\%\right]=2*\frac{\mathrm{Recall}*\mathrm{Precision}}{\mathrm{Recall}+\mathrm{Precision}}\times 100\%$$
where TP = True Positives, TN = True Negatives, FP = False Positives, and FN = False Negatives.

### 5.3. Real Multi-Resident Scenario

The proposed multi-resident activity detection sensor network has been tested in a real multi-resident scenario. The participants involved in the test were volunteers, gave written informed consent to use their personal data and were duly informed about the goal of the research. The conducted study fully respects and promotes the values of freedom, autonomy, integrity and dignity of the person, social solidarity and justice, including fairness of access. The study was carried out in compliance with the principles laid down in the Declaration of Helsinki, in accordance with the Guidelines for Good Clinical Practice.

The sensor network has been installed inside a home where two older people live: a 72-year-old man and a 64-year-old woman. A PIR sensor and a Raspberry Pi Zero W device were installed in the kitchen, bathroom, and bedroom. Figure 4 shows the floor plan of the house where the positions of the PIR sensors inside the rooms (green for the bathroom, yellow for the bedroom, blue for the kitchen) are represented by the colored points, while their detection ranges are represented by the colored areas. The PIR sensors were installed at a height of 1.5 m from the ground to better detect the movements of the residents. The walls that divide the internal rooms of the house have a thickness of 15 cm and are made of perforated brick blocks with lime plaster on both sides.

The test lasted a week, during which the residents carried out their daily activities always wearing the tag around their neck. The purpose of this test was the evaluation of the multi-resident activity detection sensor network functionality and the effectiveness of the proposed metrics in determining the AL and habits of older residents.

## 6. Results

This section presents the results of the test phases showing the accuracy of the system in detecting movements and discriminating residents, and the analysis of the data collected in the real multi-resident scenario test to assess ALs and habits of the older people.

### 6.1. Accuracy in Movement Detection

Table 4 shows the accuracy of the system in detecting small and large movements at different distances from the PIR sensor. With small movements, we refer to movements such as turning the head or moving the hands, while for large movements we refer to walking, moving arms and/or legs.

### 6.2. Accuracy in Residents Discrimination

The presented multi-resident activity detection sensor network achieved an accuracy in discriminating the identity of residents during the supervised test of 76%. Table 5 reports the precision, recall, F1-score, and accuracy of the DT, SVM, k-NN, and NB ML algorithms tested on the test dataset, split into 70% for training and 30% for testing. DT achieved 96% accuracy, 100% precision, 92% recall and 96% F1-score. SVM achieved 46% accuracy, 0% precision and recall, 63% F1-score. k-NN achieved 53% accuracy, 53% precision, 100% recall and 70% F1-score. Finally, NB achieved 65% accuracy, 86% precision, 42% recall, and 63% F1-score. Using 10-Fold CV on the test dataset, DT achieved 89% accuracy, SVM 53% accuracy, k-NN 85% accuracy and NB 66% accuracy (as reported in Table 6).

### 6.3. Real Multi-Resident Scenario Test

The data collected by the system during the test carried out in the real multi-resident scenario with the two older residents living at home were processed following the metrics proposed in Section 4.1 and Section 4.2. The data collected consist of the events detected by the system in the three rooms where the PIR motion sensors have been installed (bathroom, kitchen, bedroom), the identity of the resident associated with them, and the time of detection. The data were collected for a week, starting from 00:00 on 10 February 2022 until 00:00 on 17 February 2022. Figure 5 shows the number of detected events per day for each resident.

From the number of detected events, the ALs and the corresponding DAIs are defined for each resident, shown in Table 7.

Figure 6 shows an example of residents 1 (in blue) and 2 (in orange) HAPs for one day (12 February 2022), while Figure 7 reports the weekly HAPs average of residents 1 (in blue) and 2 (in orange). The HAPs for the entire week of both residents have been calculated.

Figure 8 and Figure 9 show the heatmaps of the HAPs for residents 1 and 2, respectively. For each hour of the days of the entire test week, the percentage of activity carried out by the residents is represented by a color. Light colors (yellow) reflect a low percentage of activity, while dark colors (blue) indicate high activity.

Figure 10 and Figure 11 show an example of the number of detected events per hour on 12 February 2022 that residents 1 and 2 carried out in each location.

The number of events captured per location has been used to compute the DRA for each resident. Figure 12 and Figure 13 report residents 1 and 2 DRA for the three rooms for each day of the week, while Figure 14 shows the weekly DRA average for residents 1 and 2.

## 7. Discussion

To assess the robustness of our system, we measured its accuracy in detecting different types of movements and discriminating against residents, and we tested it in a real-life multi-resident scenario with two older people living in their home. We then analyzed the data collected during the monitoring of the two residents through the metrics proposed in Section 4 to extract their ALs and habits.

Table 4 shows the accuracy of the system in detecting small and large movements at different distances: from a sensor-to-person distance of 1 m up to 7 m (i.e., the detection range of the system’s PIR sensors) with intermediate steps of 1 m. At a sensor-to-person distance of 1 m, detection accuracy for both small and large movements is 100%. As the distance from the sensor increases, we can observe how the system drastically decreases its ability to detect small movements, while still being able to detect large ones, although with lower accuracy. Consider that at a sensor-to-person distance of 5 m, the detection accuracy of small movements is 0% while the detection accuracy of large movements is half (50%) of that at 1 m distance (100%). These results state that the closer the person is to the PIR sensor, the better the movement is detected. If a person is sitting on a chair reading a book, the system can detect that he or she is turning the pages of the book (hand movement is considered a small movement) only if the person is close enough to the PIR sensor (we have 80% accuracy in detecting small movements at 2 m). In this case, the system detects a movement and assigns it to the person; vice versa, if the person is too far from the sensor, these small movements are not detected and the person is not assigned any activity. In the deployment of PIR sensors, it is therefore important to consider the installation in optimal points of the room to avoid losing small movements; for example, the PIR sensor can be installed near the sofa where there is a high probability of having small movements, but always taking care to cover the remaining area of the room.

During the supervised test, the multi-resident activity detection sensor network achieved an accuracy of 76% in discriminating the involved subjects. The results reported in Table 5 of the DT, SVM, k-NN and NB ML algorithms tested on the dataset, split into 70% for training and 30% for testing show that the DT classifier achieved the highest accuracy (96%), precision (100%), recall (92%), and F1-score (96%). The SVM, k-NN and NB algorithms instead prove not to be useful for improving the accuracy in discriminating residents as they achieved an accuracy of 46%, 53% and 65%, respectively. DT is one of the most widely used supervised classification algorithms. It requires no preprocessing steps and takes little time to process data compared to the other algorithms. It creates a training model that predicts the class or value of the target variable by learning simple decision rules deduced from previous data (training data). Unlike SVM, which works better on large and complex datasets, DT provides good predictions for datasets that are not composed of complex attributes, as in our case. DT algorithm is also more accurate in classification problems compared to k-NN and NB as reported in [68] and [69] respectively. However, as happens for most of the ML models, the DT classifier can be affected by overfitting problems. It is not completely possible to avoid overfitting, but with some techniques it can be mitigated. One of these is to perform a k-Fold CV on the tested dataset, which is also useful to estimate the skill of the ML model. Thus, we executed a 10-fold CV on the dataset so that the tested ML models split the data into ten sections to perform the fitting procedure a total of ten times, with each fit being performed on a training set consisting of 90% of the total training set selected at random, with the remaining 10% used as a validation set. The results reported in Table 6 show that the DT classifier also achieved the highest accuracy (89%) in this case, while SVM, k-NN and NB achieved an accuracy of 53%, 85% and 66%, respectively. Given the high accuracy of the DT classifier achieved in post-processing in the discrimination of residents, its integration in real-time data processing will be evaluated in future works.

The main aim of the test carried out in the real multi-resident scenario was the evaluation of the functionality of the multi-resident activity detection sensor network and the effectiveness of the proposed metrics in determining the AL and habits of older residents. The setup was trialed for 1 week. The collected data consisted of the events detected by the system in the three rooms where the PIR motion sensors were installed (bathroom, kitchen, bedroom), the identity of the resident, and the time of event detection. The first metric considered in the assessment of the AL of the older residents is the number of detected events (PIR sensor activation) per day. Table 7 shows the number of detected events for residents 1 and 2, thus the ALs and the related DAIs. The number of events captured during each day of the test week reflects how much active the residents were. Both residents carried out activities that classified them in MAL and HAL, thus obtaining DAIs of 2 and 3. Resident 1 achieved a DAI of 2 for 4 days and a DAI of 3 for the remaining 3 days of the week, while resident 2 achieved a DAI of 2 for 3 days and a DAI of 3 for the remaining 4 days. Resident 2 was, therefore, more active than resident 1 on a particular day (15 February 2022), while for the rest of the week both had comparable activity. We can thus say that both residents kept active during the week, having medium and high ALs.

Since the residents carry out their activities at different times of the day, the HAPs for the whole week have been calculated. Considering Figure 6 relating to the HAPs of residents 1 and 2 on 12 February 2022, the time bands of greatest activity are those between 10:00 and 13:00 and between 15:00 and 21:00. Figure 10 and Figure 11, which show the number of events detected on the same day for residents 1 and 2 in the three monitored rooms for each hour, help to better understand which kind of activity the residents were carrying out. Resident 1 is assumed to be cooking/eating and carrying out personal hygiene activities between 11.00 and 13.00 due to high activity in the kitchen and bathroom. We observe in Figure 6 a peak of activity around 12.00, which Figure 10 shows was mainly carried out in the bathroom. In the afternoon, most of the activities were detected in the kitchen, so we can assume that the resident was watching TV/relaxing on the sofa and/or eating, while around dinner time the actions can be associated with cooking/eating and personal hygiene due to events detected in the kitchen and bathroom. On the other hand, we assume that resident 2 carried out cooking/eating and watching TV/relaxing activities between 10.00 and 21.00 due to the continuous presence detected in the kitchen. We observe in Figure 6 a peak of activity around 17.00 that Figure 11 helps to relate with personal hygiene activities (probably, resident 2 took a shower).

Information on residents’ activities during the day can also be evaluated through the heatmaps shown in Figure 8 and Figure 9. The percentage of activity carried out by the residents during the days of the test week is represented by a color: a light color reflects a low percentage of activity, while a dark one reflects high activity. The heatmap thus offers a quick and easy way to understand the times of the day when older people have been most active.

Figure 7 shows the weekly HAPs average for residents 1 (in blue) and 2 (in orange). Having a weekly average of the activities carried out at various times of the day, we can have a clear picture of the periods of greatest activity of the older people: on average, both residents are most active between 10.00 and 13.00 and between 16.00 and 21.00. Contrary to what can happen for a single day (see Figure 6) in which the activities carried out may differ, from the average over a longer monitoring period (weekly in the case of the test), similar activity patterns emerged between the two residents. This may be because the couple carried out most of the activities together at home during the monitoring period, thus also bringing out behavioral habits of the couple, as well as of each individual resident.

Figure 12 and Figure 13 show residents 1 and 2 DRA for the test week, while Figure 14 shows the weekly DRA average for both residents. Most of the activity carried out by residents was in the kitchen, given the high value of average DRA in it (74.80% for resident 1 and 75.60% for resident 2). The average DRA of the bathroom is 18.40% for resident 1 and 18.90% for resident 2, while we notice a low DRA value for the bedroom (6.80% for resident 1 and 5.50% for resident 2), which can be explained by the fact that the PIR sensor is only activated when the residents enter/leave the room or dress in it. Most of the time spent in the bedroom is in fact associated with the action of sleeping, so we expected low values of activity detected in it, resulting in low DRA values.

## 8. Conclusions

The main contribution of this work is the presentation of an innovative multi-resident activity detection sensor network that uses the BLE signal emitted by tags worn by residents and PIR motion sensors deployed in the house to locate residents and monitor their activities. The advantage of this system is that it guarantees remote monitoring, is non-invasive, low-cost, and respectful of privacy. Furthermore, this system can be easily extended to larger environments and contexts with more residents. Initially designed to be installed in the home of older people, the sensor network can also be used in nursing homes where the monitoring of older residents can help caregivers understand their habits and daily activities. By integrating the system with more Raspberry Pi Zero W devices and PIR motion sensors, we can cover more areas, and by using additional tags, we can track more people. A drawback of using BLE technology is that radio waves are influenced by the presence of external factors such as walls, objects, etc. (generate absorption, interference, diffraction), which cause the RSSI to fluctuate. The further away the tag is from the Raspberry Pi Zero W scanner, the more unstable the RSSI is and the less accurate the detection becomes. Another drawback of using the wearable approach can be forgetting to wear the tag. In this case, the sensor network can still detect the movements of the users, but it is not able to associate them with the users’ identities.

We demonstrated that this system solves the problem of monitoring older people and measuring their activities in multi-resident contexts through a test carried out in a real multi-resident scenario where two older people were monitored during their daily life. Finally, the metrics defined to analyze and interpret the collected data revealed a useful and innovative approach to understand daily habits and measure the AL of older people: the number of detected events (PIR activations) reflects the activity level of the older people, while the number of events captured per location reflects the amount of activity performed by the older people assigned to execute a specific ADL. By analyzing the results obtained, it is thus possible to define a behavioral model for the older people. The assessment of this model can help to capture possible changes resulting from cognitive or physical health.

For future works, a case will be designed to house each PIR motion sensor and the related Raspberry Pi Zero W. Furthermore, the use of wireless PIR sensors instead of wired ones will be evaluated in order to reduce the number of Raspberry Pi Zero W devices by connecting more than one PIR to each of them, thus reducing costs and eliminating wiring. We found that the proposed sensor network can achieve high accuracy in discriminating the identity of residents by using the DT classifier; thus, further investigation regarding its real-time application will be carried out.

## Figures and Tables

**Figure 1 sensors-22-03472-f001:**
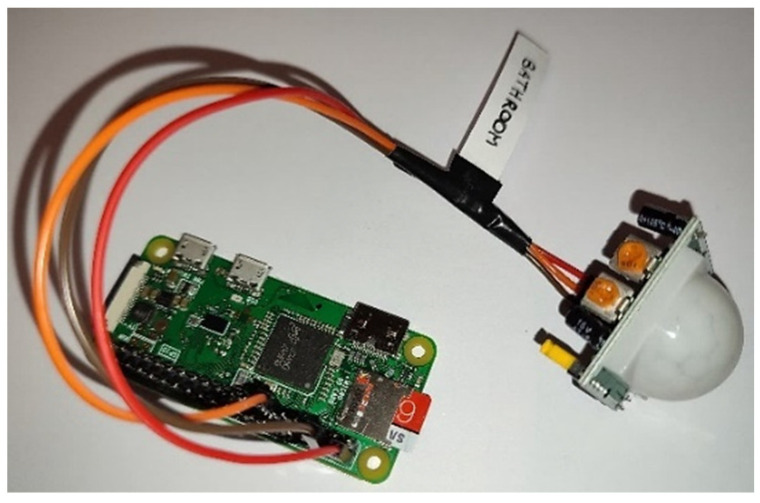
HC-SR501 PIR sensor wired to the Raspberry Pi Zero W. The red wire connects the Vcc pin of the PIR sensor to the 5 V pin (PIN 2) of the Raspberry; the brown wire connects the Gnd pin of the PIR sensor to the ground pin (PIN 6) of the Raspberry; the orange wire connects the Out pin of the PIR sensor to a GPIO pin (PIN 13) of the Raspberry.

**Figure 2 sensors-22-03472-f002:**
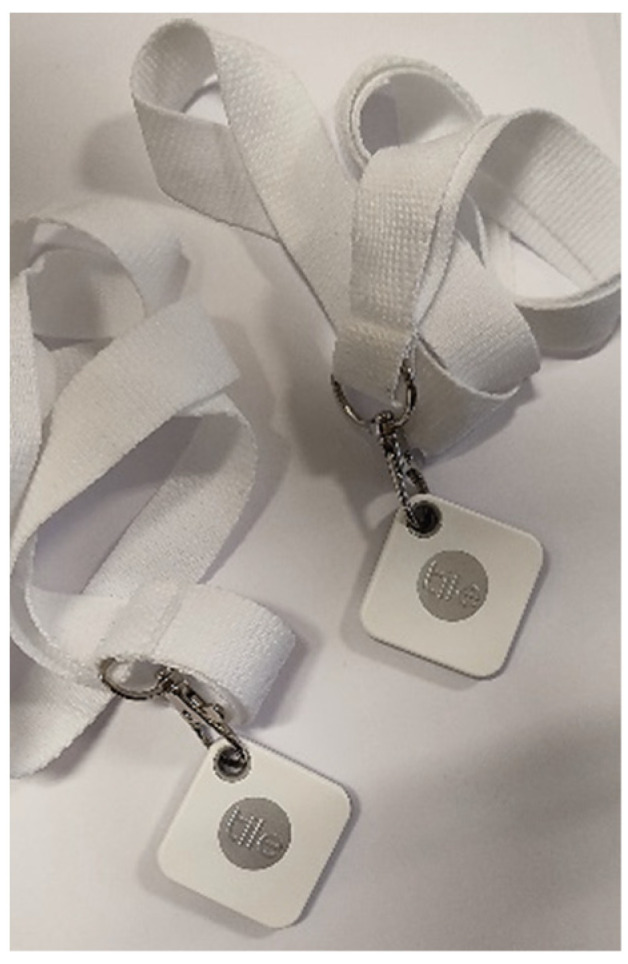
Tile Mate tags.

**Figure 3 sensors-22-03472-f003:**
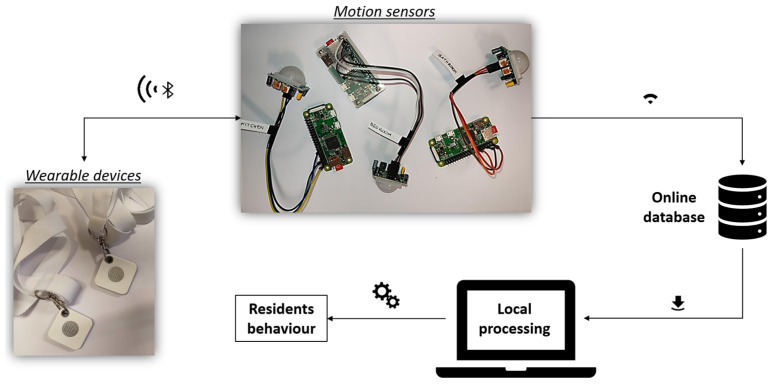
Multi-resident detection system setup.

**Figure 4 sensors-22-03472-f004:**
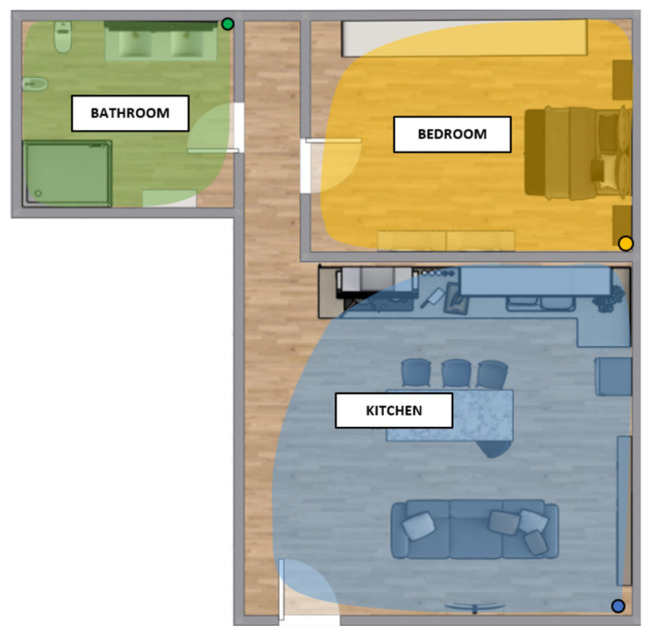
Floor plan of the real home scenario. The colored areas represent the detection ranges of each PIR sensor, which are represented by the colored points.

**Figure 5 sensors-22-03472-f005:**
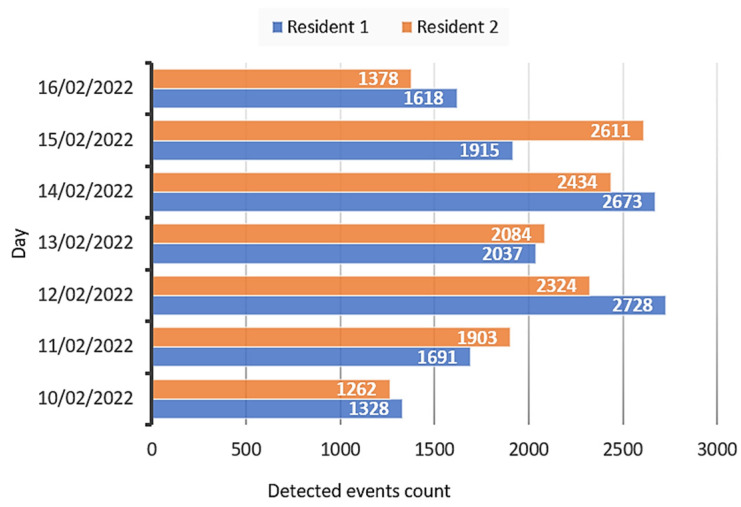
Detected events count per day for resident 1 (in blue) and resident 2 (in orange).

**Figure 6 sensors-22-03472-f006:**
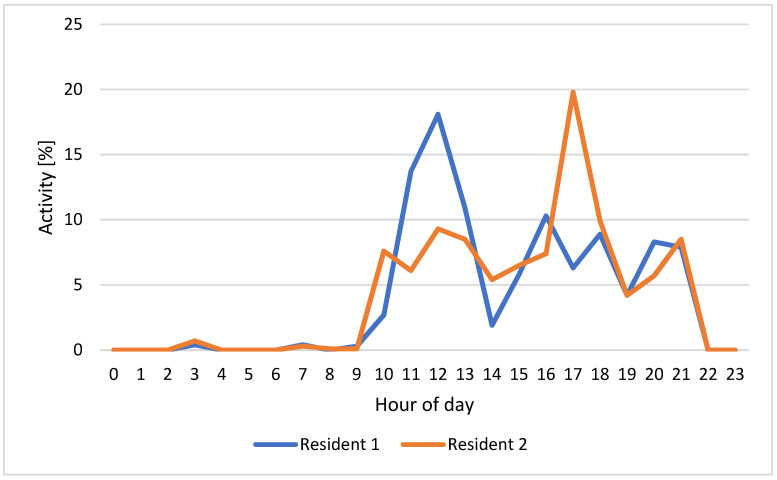
HAP for residents 1 (in blue) and 2 (in orange) on 12 February 2022.

**Figure 7 sensors-22-03472-f007:**
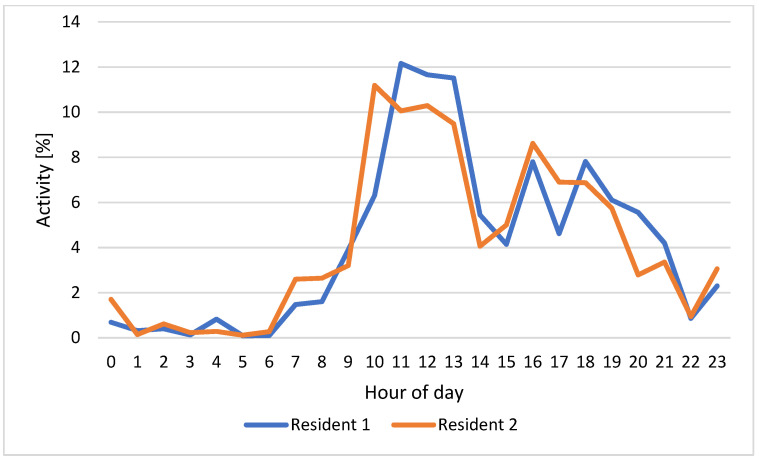
Weekly HAPs average for residents 1 (in blue) and 2 (in orange).

**Figure 8 sensors-22-03472-f008:**
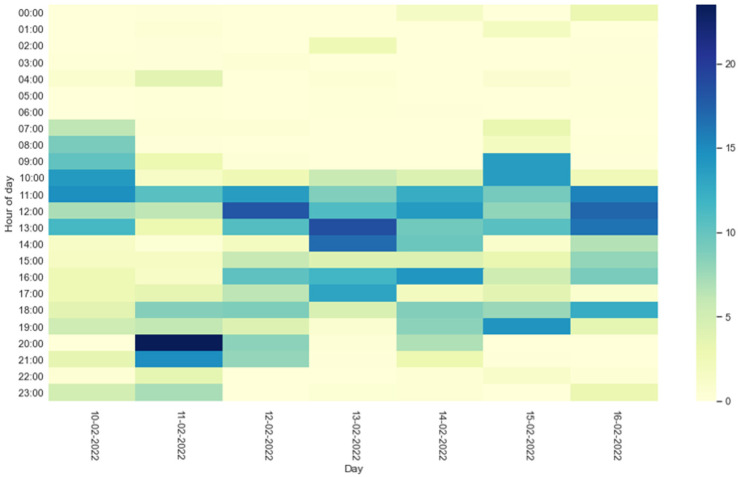
Heatmap of resident 1 HAPs during the test week. The legend indicates the color relating to the percentage of activity in a given hour.

**Figure 9 sensors-22-03472-f009:**
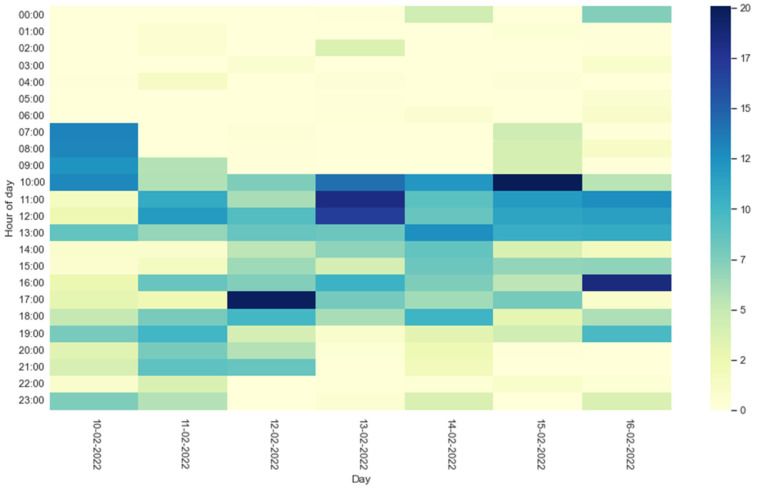
Heatmap of resident 2 HAPs during the test week. The legend indicates the color relating to the percentage of activity in a given hour.

**Figure 10 sensors-22-03472-f010:**
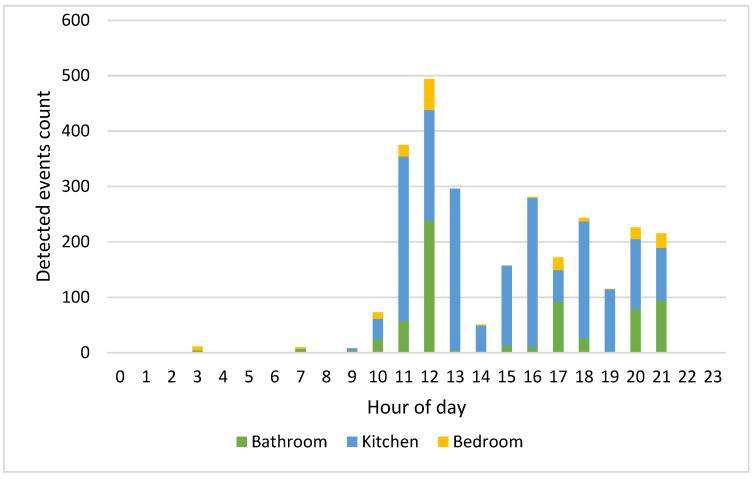
Resident 1 detected events per hour on 12 February 2022 in each room.

**Figure 11 sensors-22-03472-f011:**
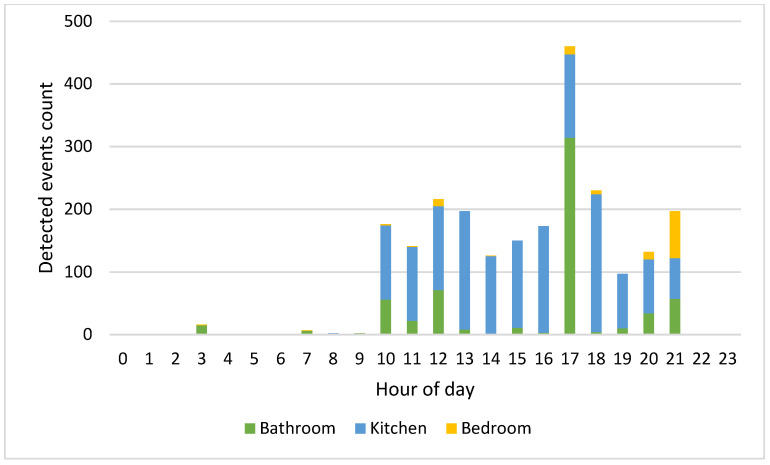
Resident 2 detected events per hour on 12 February 2022 in each room.

**Figure 12 sensors-22-03472-f012:**
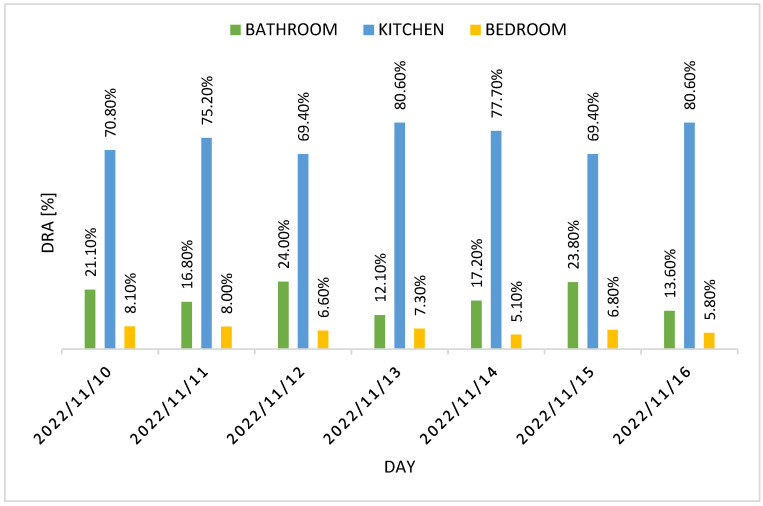
Resident 1 DRA for each day of the test week.

**Figure 13 sensors-22-03472-f013:**
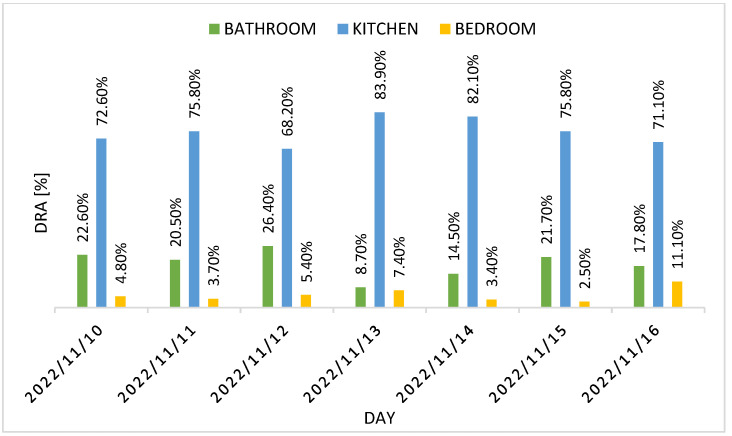
Resident 2 DRA for each day of the test week.

**Figure 14 sensors-22-03472-f014:**
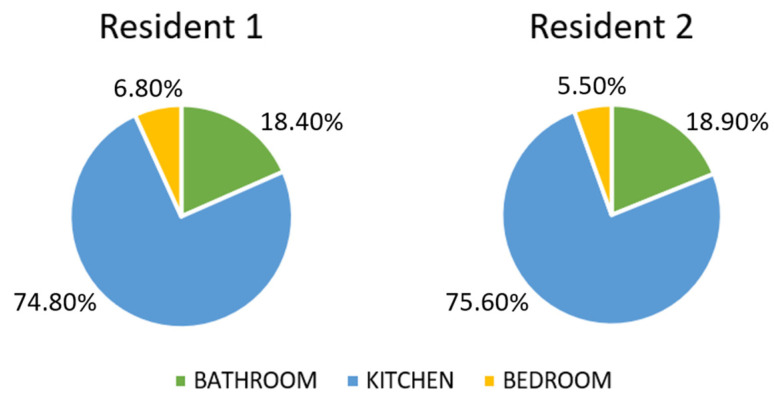
Weekly DRA average for residents 1 and 2.

**Table 1 sensors-22-03472-t001:** HC-SR501 PIR sensor technical features.

Features	PIR Sensor (HC-SR501)
Detection range	7 m
Detection angle	120 degrees
Delay time	3 s
Consumption	65 mA
Operating voltage	4–12 V (5 V recommended)
Output voltage	3.3 V
Operating temperature	−15–70 °C
Cost	EUR 3

**Table 2 sensors-22-03472-t002:** Tile Mate technical features.

Features	Tile Mate BLE Tag
Dimensions	37.8 mm × 37.8 mm × 7.1 mm
Range	76 m
Battery	Up to 3-year battery life
Water-resistant	Water-resistant (IP67)
Cost	EUR 25

**Table 3 sensors-22-03472-t003:** Raspberry Pi Zero W technical features.

Features	Raspberry Pi Zero W
Communication technology	Wi-Fi 802.11 b/g/n, Bluetooth 4.1, BLE
CPU	Single-core 1 GHz
RAM	512 MB
Communication ports	Mini HDMI, micro–USB OTG, micro–USB power, HAT-compatible 40-pin header, composite video and reset headers, CSI camera connector
Cost	EUR 25

**Table 4 sensors-22-03472-t004:** Accuracy of the multi-resident activity detection sensor network in recognizing small and large movements at seven different sensor-to-person distances.

Sensor-to-Person Distance [m]	Movement	Accuracy
1	Small	100%
	Large	100%
2	Small	80%
	Large	100%
3	Small	10%
	Large	80%
4	Small	5%
	Large	60%
5	Small	0%
	Large	50%
6	Small	0%
	Large	40%
7	Small	0%
	Large	25%

**Table 5 sensors-22-03472-t005:** Precision, recall, F1-score, and accuracy in percentage values of the DT, SVM, k-NN, and NB algorithms applied to the supervised test dataset split into 70% for training and 30% for testing.

ML Algorithms	Precision [%]	Recall [%]	F1-Score [%]	Accuracy [%]
DT	100	92	96	96
SVM	0	0	63	46
k-NN	53	100	70	53
NB	85	42	63	65

**Table 6 sensors-22-03472-t006:** Accuracy in percentage values of the DT, SVM, k-NN, and NB algorithms applied to the supervised test dataset split by using 10-Fold CV.

ML Algorithms	Accuracy [%]
DT	89
SVM	53
k-NN	85
NB	66

**Table 7 sensors-22-03472-t007:** Detected events count, AL and DAI for each resident on each day of the test.

Day	Identity	Detected Events Count	AL	DAI
10 February 2022	Resident 1	1328	MAL	2
	Resident 2	1262	MAL	2
11 February 2022	Resident 1	1691	MAL	2
	Resident 2	1903	MAL	2
12 February 2022	Resident 1	2728	HAL	3
	Resident 2	2324	HAL	3
13 February 2022	Resident 1	2037	HAL	3
	Resident 2	2084	HAL	3
14 February 2022	Resident 1	2673	HAL	3
	Resident 2	2434	HAL	3
15 February 2022	Resident 1	1915	MAL	2
	Resident 2	2611	HAL	3
16 February 2022	Resident 1	1618	MAL	2
	Resident 2	1378	MAL	2

## Data Availability

Not applicable.

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
