# Peer review of "The Problem of Monitoring Activities of Older People in Multi-Resident Scenarios: An Innovative and Non-Invasive Measurement System Based on Wearables and PIR Sensors"

_sensors, 2022, doi:10.3390/s22093472_

Round 1

Reviewer 1 Report

The paper propose a solution to monitor the daily activities of 2 elderly people which share the same appartment. The hardware design and software treatment of the solution are very well explained. The proposed measures and the floor plan  are presented. The wall type can be interesting to be indicated because is important for RSSI measure.

The scenarized stage is used to evaluate the sensor system. The author must mention the test number for each position for IR and also for ID detection. How these data was labelled ? 

The resident discrimination is made on 70/30 data base division but a K-Fold (5 or 10) is more interesting statistically.

For the 1 week recording analysis, the author don't have any reference information about the real activity of the people ? Can be interesting to use video camera in order to evaluate the system for a long period : 1 week).

The possibility to install 2 or more PIR sensor on the same Raspberry Pi can allow better results with a small price (IR sensor is only 3 euros) ?

Reviewer 2 Report

In this paper, the author proposed an innovative multi-resident activity detection sensor network that uses the Bluetooth Low Energy signal emitted by tags worn by residents and passive infrared (PIR) motion sensors deployed in the house to locate residents and monitor their activities. The results are corroborative and useful. I can recommend it for a publication after several concerns are properly addressed.  

  1. The authors should rephrase their main contributions or advantages of this method in the introduction part.

  1. The literature review part is weak. The authors should rephrase their Section II with a more clear logic. Several summarizing sentences should be inserted after paragraph in order to strength potential research motivative.

  1. Why the DT classifier performs the best? Can the authors provide a proper analysis?

  1. The authors should provide detailed advantages or disadvantages for the three most used solutions including vision-based, tag-based and signature-based techniques.
